A cost-effective predictive tool for AFP-negative focal hepatic lesions of retrospective study: enhancing clinical triage and decision-making

Lin Yu 1
Wang Qianyi 2
Feng Minxuan 1
Lao Jize 1
Wu Changmeng 1
Luo Houlong 1
Ji Ling 1 1120303921@qq.com
Xia Yong 1 sunmoonrain78@163.com
1 Department of Laboratory Medicine, Peking University Shenzhen Hospital , Shenzhen, Guangdong Province , China
2 Department of Laboratory Medicine, JingNing People’s Hospital , Pingliang, Gansu Province , China
Oliveira Sonia
Electronic publication date: 2025 Mar 26
Publication date: 2025
Volume: 13
Electronic Location ID: e19150
Received 2024 Nov 1; Accepted 2025 Feb 19
Copyright: © 2025 Lin et al.
Copyright year: 2025
Copyright holder: Lin et al.
License: This is an open access article distributed under the terms of the Creative Commons Attribution License, which permits unrestricted use, distribution, reproduction and adaptation in any medium and for any purpose provided that it is properly attributed. For attribution, the original author(s), title, publication source (PeerJ) and either DOI or URL of the article must be cited.
License URL: https://creativecommons.org/licenses/by/4.0/

Keywords: Alpha-fetal protein (AFP), Focal hepatic lesions, AFP-negative, Diagnostic model, Multivariate logistic regression, Clinical validation

Funding: Shenzhen Science and Technology Program JCYJ20220531093816038 This work was supported by Shenzhen Science and Technology Program (JCYJ20220531093816038). The funders had no role in study design, data collection and analysis, decision to publish, or preparation of the manuscript.

==============================
Background

Identifying alpha-fetal protein (AFP)-negative focal hepatic lesions presents a significant challenge, particularly in China. We sought to develop an economically portable tool for the diagnosis of benign and malignant liver lesions with AFP-negative status, and explore its clinical diagnostic efficiency.

Methods

A retrospective study was conducted at Peking University Shenzhen Hospital from January 2017 to February 2023, including a total of 348 inpatients with AFP-negative liver space-occupying lesions. The study used a training set of 252 inpatients from January 2017 to September 2021 to establish a diagnostic model for differentiating benign and malignant AFP-negative liver space-occupying lesions. Additionally, a validation cohort of 96 inpatients from October 2021 to February 2023 was used to confirm the diagnostic performance of the model. From January 2017 to February 2023, patients at JingNing People’s Hospital, Gansu Province were assigned to the external cohort (n = 78).

Results

A predictive tool was established by screening age, gender, hepatitis B virus (HBV)/hepatitis C virus (HCV) infected, single lesion, alanine amino transferase (ALT), and lymphocyte-to-monocyte ratio (LMR) using multivariate logistic regression analysis and clinical practice. The area under the curve (AUC) of the model was 0.911 (95% CI [0.873–0.949]) in the training set and 0.882 (95% CI [0.815–0.949]) in the validation cohort. In addition, the model achieved an area under the curve of 0.811 (95% CI [0.687–0.935]) in the external validation cohort.

Conclusion

Our results demonstrated that the predictive tool has the characteristics of good diagnostic efficiency, economy and convenience, which is helpful for the clinical triage and decision-making of AFP-negative liver space-occupying lesions.

Introduction

With the advancement and widespread application of diagnostic imaging, there has been a gradual increase in the detection rate of focal hepatic lesions. These lesions, categorized as benign or malignant based on their invasiveness, predominantly comprise primary liver cancer, contributing to approximately 80% of malignant cases and ranking as the third leading cause of global cancer-related mortality (Forner, Reig & Bruix, 2018; Llovet et al., 2018). However, early-stage diagnosis of liver cancer occurs in less than 40% of patients (Han et al., 2014). Timely discrimination and early intervention or treatment for patients with focal hepatic lesions hold significant clinical relevance. Approximately one-third of hepatocellular carcinoma (HCC) patients exhibit serum alpha-fetal protein (AFP) negativity (Zhang et al., 2018), which is very common in cirrhotic patients (Ma, Wang & Teng, 2013), with small liver tumors being particularly prevalent in this group. AFP-negative HCC is often characterized by poor differentiation, which is associated with a worse prognosis. This subtype poses unique diagnostic challenges, complicating early detection and treatment.

While determining the nature of focal hepatic lesions is pivotal in clinical practice, radiological assessment presents challenges. The discrimination heavily relies on the expertise of radiologists, introducing subjectivity, and focal lesions may lack typical radiological features. In cases where radiological characterization is challenging, AFP serves as an effective auxiliary tool (Llovet et al., 2021). AFP, a glycoprotein widely utilized in diagnosing and monitoring liver cancer, exhibits a sensitivity of only 60% at the optimal threshold (10–20 ng/mL) for HCC, with limited specificity (Jing et al., 2020; Trevisani et al., 2001). Reports indicate that up to 30% of HCC patients exhibit negative serum AFP levels (Luo et al., 2019). In China, most people are screened by non-specific indicators in physical examination and imaging. After the detection of liver-occupying lesions, clinical management is faced with a dilemma: to continue conservative observation or to resort to invasive investigations to make a definitive diagnosis. Further clinical diagnosis of malignancy or benignity of the lesion becomes challenging, especially in patients with negative AFP levels. Therefore, the establishment of a reliable tool to accurately predict and identify the malignancy or benignity of AFP-negative focal hepatic lesions to achieve clinical triage, is of great value in guiding clinical decision-making.

Previously, various clinical prediction models, such as GALAD (Johnson et al., 2014) and ASAP (Yang et al., 2019), have been employed for liver cancer discrimination and diagnosis, with AFP playing a crucial role despite their excellent performance. In contrast, staging systems like AJCC (Amin et al., 2017), BCLC (Llovet, Brú & Bruix, 1999), JIS score (Kudo, Chung & Osaki, 2003) and ALBI grade (Johnson et al., 2015) are not apt for distinguishing the malignancy or benignity of focal hepatic lesions. Recent studies have also attempted to construct clinical prediction models for AFP-negative liver cancer (Liu et al., 2023; Luo et al., 2019). However, the control groups in these studies only included healthy individuals, neglecting the fact that the patient population encountered by physicians in actual clinical practice is diverse, potentially including patients with other benign focal hepatic lesions.

In the actual clinical environment, the most challenging aspect of diagnosing AFP-negative HCC is excluding patients with benign lesions. The economic level of different regions in China varies greatly, so the economy and convenience of the model are also of particular importance. Given these challenges—the significant proportion of overlooked AFP-negative liver cancer cases, the dilemma of clinical triage and decision-making for AFP-negative focal hepatic lesions, and the inadequacy of existing prediction models to fully meet clinical needs—our objective is to establish a reliable predictive tool for discriminating and diagnosing AFP-negative focal hepatic lesions. This predictive tool is designed to be straightforward, cost-effective, and clinically practical.

Materials and Methods

Ethical considerations

This study was conducted with the approval of the Ethics Committees of Peking University Shenzhen Hospital (Peking University Shenzhen Hospital Ethical Approval [Research] [2023] No. 075) and JingNing People’s Hospital, Gansu Province (Jingning People’s Hospital Ethical Approval No. 2023101501). Due to the retrospective nature of the analysis, a waiver of informed consent was granted by both Ethics Committees. All processes were carried out in a de-identified manner to protect patient privacy, ensuring confidentiality throughout the study.

We certify that the study was performed in accordance with the 2013 Declaration of Helsinki, upholding ethical standards in research involving human subjects. By addressing these ethical considerations, we reinforce the credibility of our study and the integrity of our findings.

Participants

A retrospective study was conducted at Peking University Shenzhen Hospital from January 2017 to February 2023, including a total of 348 inpatients with AFP-negative liver space-occupying lesions. Patients were included if they had a confirmed diagnosis of liver lesions through imaging and histopathological examination. Exclusion criteria consisted of patients with prior liver resection, incomplete medical records, or lesions with positive AFP status. The study utilized a training set of 252 inpatients from January 2017 to September 2021 to establish a diagnostic model for differentiating benign and malignant AFP-negative liver space-occupying lesions. Data were collected through electronic medical records, ensuring comprehensive patient information on demographics, laboratory results, and clinical history. Additionally, a validation cohort of 96 inpatients from October 2021 to February 2023 was used to confirm the diagnostic performance of the model. From January 2017 to February 2023, patients at JingNing People’s Hospital, Gansu Province were assigned to the external cohort (n = 78) (Fig. 1).

Figure 1 Flowchart of the study.

Data collection

In this study, the medical records were checked using the hospital information system (HIS). Clinical data, including the patient’s personal characteristics, medical history, imaging examination and pathology reports, were collected. Relevant laboratory test data were collected from the laboratory information system (LIS). All data utilized in this study are sourced from pre-therapy cross-sections. AFP levels were detected using Abbott with a threshold of 20 ng/mL, below which a diagnosis of AFP-negative hepatocellular carcinoma (HCC) was established.

The following information should be included, (1) personal characteristics, gender, age, and drinking history; (2) medical history, history of viral hepatitis, liver cirrhosis, diabetes, malignant tumors, family history of cancer, and non-alcoholic fatty liver disease; (3) imaging data, number of lesions; (4) pathology report, nature of liver space-occupying lesions; (5) laboratory tests, liver function tests including alanine amino transferase (ALT), total protein (TP), albumin (ALB) and Direct bilirubin (DB); blood routine tests including white blood cells (WBC), neutrophil percentage (NEUT%), lymphocyte percentage (LYMPH%), mononuclear cell ratio (MONO%), red blood cells (RBC), hemoglobin (Hb), hematocrit (HCT), mean corpuscular volume (MCV), mean corpuscular hemoglobin concentration (MCHC), mean hemoglobin volume (MCH), red blood cell standard deviation of distribution width (RDW-SD), coefficient of variation of red blood cell size (RDW-CV), platelet (PLT), mean platelet volume (MPV), and platelet distribution width (PDW); calculation items such as neutrophil calculated ratio of lymphocytes to neutrophils (NLR), calculated ratio of platelets to lymphocytes (PLR), and calculated ratio of lymphocytes to monocytes (LMR).

Statistical analysis

To develop the predictive model, we employed multivariate logistic regression analysis due to its ability to handle multiple independent variables while estimating the probability of a binary outcome—benign or malignant lesions. This method allows for the assessment of the influence of each variable on the outcome, providing insights into the relative importance of different predictors. Model selection was guided by the Akaike information criterion (AIC), ensuring optimal fit without overfitting. Cross-validation techniques were applied to assess the model’s robustness, reducing the risk of biased results. The model’s diagnostic performance was evaluated using the area under the curve (AUC) from receiver operating characteristic (ROC) analyses, with a higher AUC indicating better discrimination between benign and malignant lesions. Statistical significance was determined at a p-value of less than 0.05, and all analyses were performed using R software on Windows and the Beckman Coulter DxAI platform (https://dxonline.deepwise.com).

Results

Patients characteristics

According to the inclusion and exclusion criteria, a total of 426 patients were included in the study, including 289 (67.84%) patients in the benign group and 137 (32.16%) in the malignant group. Benign liver space-occupying lesions commonly seen include hepatic hemangioma (HCH), hepatic cyst, focal nodular hyperplasia of the liver (FNH), liver abscess (DLA), hepatic adenoma (HCA), and inflammatory pseudotumor of the liver (IPL). On the other hand, malignant liver space-occupying lesions primarily consist of primary liver cancer, such as hepatocellular carcinoma (HCC), intrahepatic cholangiocarcinoma (ICC), and mixed hepatocellular carcinoma-cholangiocarcinoma (cHCC-ICC), as well as liver metastases (ML) and hepatic sarcoma. There was no significant difference in the general clinical information between the training sets and the internal validation sets (P > 0.05), indicating that the grouping was randomized and reasonable, while the external validation had some difference. More baseline information is shown in Table 1.

Table 1 Demographics and clinical characteristics of training and validation sets.

Characteristics	Training cohort	Internal validation cohort	External validation cohort	
Benign
(n = 164)	Malignant
(n = 88)	Benign
(n = 60)	Malignant
(n = 36)	Benign
(n = 65)	Malignant
(n = 13)	
Age	47.17 ± 13.75	58.18 ± 12.33	47.82 ± 15.41	57.42 ± 11.65	58.37 ± 12.35*	62.54 ± 7.43	
Sex							
Male	79 (48.20%)	68 (77.30%)	21 (35.00%)	30 (83.30%)	26 (40.00%)	8 (61.50%)	
Female	85 (51.80%)	20 (22.70%)	39 (65.00%)	6 (16.70%)	39 (60.00%)	5 (38.50%)	
Alcohol							
Yes	17 (10.37%)	22 (25.00%)	6 (10.00%)	10 (27.80%)	2 (3.10%)	2 (15.40%)	
No	147 (89.63%)	66 (75.00%)	54 (90.00%)	26 (72.20%)	63 (96.90%)	11 (84.60%)	
Steatosis or Cirrhosis							
Yes	7 (4.27%)	23 (26.14%)	2 (3.30%)	12 (33.30%)	4 (6.20%)	4 (30.80%)	
No	157 (95.73%)	65 (73.86%)	58 (96.70%)	24 (66.70%)	61 (93.80%)	9 (69.20%)	
Family history							
Yes	6 (3.66%)	5 (5.68%)	2 (3.30%)	0 (0.00%)	0 (0.00%)	0 (0.00%)	
No	158 (96.34%)	83 (94.32%)	58 (96.70%)	36 (100.00%)	65 (100.00%)	13 (100.00%)	
Diabetes							
Yes	4 (2.44%)	12 (13.64%)	2 (3.30%)	7 (19.40%)	0 (0.00%)	1 (7.70%)	
No	160 (97.56%)	76 (86.36%)	58 (96.70%)	29 (80.60%)	65 (100.00%)	12 (92.30%)	
HBV infected							
Yes	17 (10.37%)	45 (51.13%)	6 (10.00%)	17 (47.22%)	6 (9.23%)	1 (7.69%)*	
No	147 (89.63%)	43 (48.86%)	54 (90.00%)	19 (52.78%)	59 (90.77%)	12 (92.31%)*	
HCV infected							
Yes	1 (0.61%)	3 (3.41%)	0 (0.00%)	2 (5.56%)	1 (1.54%)	1 (7.69%)	
No	163 (99.39%)	85 (96.59%)	60 (100.00%)	34 (94.44%)	64 (98.46%)	12 (92.31%)	
Single lesion							
Yes	45 (47.87%)	49 (52.13%)	27 (45.00%)	23 (63.90%)	14 (21.50%)	4 (30.80%)	
No	112 (78.32%)	31 (21.68%)	33 (55.00%)	13 (36.10%)	51 (78.50%)	9 (69.20%)	
ALT, U/L	22.79 ± 18.01	38.65 ± 35.06	19.5 ± 14.51*	32.97 ± 18.69	22.94 ± 15.58	49.38 ± 42.64	
TP, g/L	67.79 ± 5.25	67.20 ± 6.33	67.49 ± 4.48	68.18 ± 6.36	68.56 ± 6.63	70.21 ± 8.24	
Alb, g/L	40.31 ± 3.50	37.82 ± 4.67	39.39 ± 3.10	37.44 ± 4.88	43.91 ± 4.46*	40.80 ± 5.14	
DB, mg/dL	2.64 ± 1.98	11.58 ± 37.08	2.35 ± 0.99	4.45 ± 4.36	11.95 ± 44.38*	15.45 ± 16.18*	
WBC, E+9/L	5.98 ± 2.40	6.10 ± 2.34	5.83 ± 1.70	6.40 ± 2.68	5.93 ± 4.78	7.20 ± 2.36*	
NEUT%	55.07 ± 9.76	58.27 ± 12.54	54.23 ± 10.75	59.26 ± 11.02	60.73 ± 12.86*	67.45 ± 5.72*	
LYMPH%	33.94 ± 8.93	28.65 ± 10.52	34.26 ± 9.28	28.35 ± 10.18	30.8 ± 11.73	22.85 ± 5.92	
MONO%	7.70 ± 2.03	7.70 ± 2.03	7.95 ± 2.15	9.00 ± 1.79	6.69 ± 1.71*	8.49 ± 2.18	
RBC, E+12/L	4.52 ± 0.60	4.34 ± 0.81	4.52 ± 0.58	4.32 ± 0.85	4.67 ± 0.51*	4.23 ± 0.58	
Hb, g/L	131.21 ± 18.54	130.17 ± 21.55	129.22 ± 14.54	129.78 ± 22.02	144.62 ± 20.17*	129.15 ± 18.68	
HCT, %	40.08 ± 5.00	39.49 ± 6.19	40.01 ± 4.19	39.29 ± 6.52	42.66 ± 5.07*	38.62 ± 5.53	
MCV, fL	89.02 ± 7.49	91.66 ± 8.31	89.10 ± 7.97	91.51 ± 7.11	91.41 ± 5.73	91.31 ± 4.27	
MCHC, g/L	326.85 ± 12.92	329.09 ± 14.02	322.85 ± 11.75*	330.31 ± 12.93	338.28 ± 16.87*	335.08 ± 11.40	
MCH, pg	29.14 ± 3.02	30.22 ± 3.35	28.81 ± 3.09	30.24 ± 2.70	30.95 ± 2.72*	30.59 ± 1.89	
RDW-SD, fL	41.85 ± 3.35	44.64 ± 6.10	42.24 ± 3.96	43.80 ± 6.13	42.90 ± 4.47	43.80 ± 3.81	
RDW-CV, %	13.03 ± 1.65	13.42 ± 1.76	13.04 ± 1.18	13.25 ± 2.74	12.86 ± 1.44	13.22 ± 1.44	
PLT, E+9/L	224.99 ± 63.14	183.26 ± 81.40	237.20 ± 63.01	191.67 ± 121.89	219.09 ± 77.23	224.23 ± 74.90	
MPV, fL	10.75 ± 1.10	11.01 ± 1.16	10.36 ± 0.98	10.78 ± 1.10	10.05 ± 0.95	9.94 ± 0.83*	
PDW, fL	12.85 ± 2.66	13.34 ± 2.97	11.75 ± 2.04	12.81 ± 2.91	11.38 ± 2.22	10.63 ± 1.68*	
NLR	2.03 ± 2.30	2.62 ± 1.86	1.82 ± 0.95	2.60 ± 1.59	3.13 ± 4.08*	3.22 ± 1.16	
PLR	125.27 ± 47.76	126.59 ± 83.93	131.77 ± 46.65	119.84 ± 61.26	172.36 ± 189.14*	152.44 ± 55.14	
LMR	4.68 ± 1.76	3.37 ± 1.38	4.58 ± 1.56	3.29 ± 1.51	4.74 ± 1.78	2.91 ± 1.22	
Notes:

HBV, hepatitis B virus; HCV, hepatitis C virus; ALT, alanine aminotransferase; TP, total Protein; Alb, albumin; DB, direct bilirubin; WBC, leukocyte; NEUT, neutrophil; LYMPH, lymphocyte; MONO, monocyte; RBC, erythrocyte; Hb, hemoglobin; HCT, packed cell volume; MCV = HCT / RBC; MCHC = HB / HCT; MCH = HB / RBC; RDW-SD, the standard deviation of erythrocyte distribution width; RDW-CV, the coefficient of variation of erythrocyte distribution width; PLT, platelet; MPV, Mean platelet volume; PDW, platelet distribution width; NLR, neutrophil-to-lymphocyte ratio; PLR, platelet-to-lymphocyte ratio; LMR, lymphocyte-to-monocyte ratio.

* P ≤ 0.05

Predictor selection of malignant AFP-negative liver space-occupying lesions

Univariate and multifactorial logistic regression analyses were performed to assess the association between clinical characteristics and malignant AFP-negative liver space-occupying lesions. Univariate logistic regression analysis showed that age, sex, alcohol, history of non-alcoholic fatty liver or cirrhosis, history of diabetes, history of viral hepatitis, single lesion, ALT, ALB, total bilirubin (TB), LYMPH%, MONO%, NEUT%, MCV, MCH, RDW-SD, PLT, and LMR were identified as candidate predictors of malignant AFP-negative liver space-occupying lesions. Multi-factors logistic regression analysis revealed that age (OR: 1.08, 95% CI [1.05–1.12]), hepatitis B virus (HBV)/hepatitis C virus (HCV) infected (OR: 14.99, 95% CI [6.20–36.24]), single lesion (OR: 4.81, 95% CI [2.19–10.58]), ALT (OR: 1.03, 95% CI [1.01–1.05]) and LMR (OR: 0.68, 95% CI [0.52–0.80]) were independent predictors for malignant AFP-negative liver space-occupying lesions (Table 2).

Table 2 Univariate and multivariate logistic analysis of the training cohort.

Characteristics	Univariate analysis	Multivariate analysis	
OR (95% CI)	P value	OR (95% CI)	P value	
Age	1.06 [1.04–1.09]	<0.001	1.08 [1.05–1.12]	<0.001	
Sex	3.66 [2.04–6.57]	<0.001	1.39 [0.62–3.13]	0.427	
Alcohol	2.88 [1.44–5.78]	0.003			
Steatosis or Cirrhosis	7.94 [3.25–19.41]	<0.001			
Diabetes	6.32 [1.97–20.23]	0.003			
HBV/HCV infected	9.73 [5.11–18.55]	<0.001	14.99 [6.20–36.24]	<0.001	
Single lesion	3.32 [1.93–5.72]	<0.001	4.81 [2.19–10.58]	0.001	
ALT, U/L	1.03 [1.02–1.04]	<0.001	1.03 [1.01–1.05]	0.001	
Alb, g/L	0.85 [0.79–0.92]	<0.001			
DB, mg/dL	1.25 [1.08–1.44]	0.003			
NEUT%	1.03 [1.01–1.05]	0.028			
LYMPH%	0.94 [0.92–0.97]	<0.001			
MONO%	1.26 [1.12–1.42]	<0.001			
MCV, fL	1.05 [1.01–1.09]	0.013			
MCH, pg	1.13 [1.03–1.25]	0.012			
RDW-SD, fL	1.17 [1.09–1.25]	<0.001			
PLT, E+9/L	0.99 [0.99–1.00]	<0.001			
NLR	1.15 [0.99–1.32]	0.067			
LMR	0.57 [0.47–0.70]	<0.001	0.68 [0.52–0.80]	0.003	
Notes:

HBV, hepatitis B virus; HCV, hepatitis C virus; ALT, alanine aminotransferase; Alb, albumin; DB, direct bilirubin; NEUT, neutrophil; LYMPH, lymphocyte; MONO, monocyte; MCV = HCT/RBC; MCH = HB/RBC; RDW-SD, the standard deviation of erythrocyte distribution width; PLT, platelet; NLR, neutrophil-to-lymphocyte ratio; LMR, lymphocyte-to-monocyte ratio.

Develop and validate the nomogram

The independent predictors found by multivariate analysis were used to develop a nomogram (Fig. 2). In the training cohort, the ROC curve revealed that AUC was 0.911 (95%CI [0.873–0.949]) (Fig. 3A). The calibration curve illustrates a close match between the risk of the model diagnosis and the actual observed risk (Fig. 3B). The model prediction and actual observation of benign and malignant patients with AFP-negative liver space-occupying lesions are comparable, showing good consistency in both the training set and internal validation. To assess the clinical usability and benefits of the predictive models, clinical decision curves (DCA) were used to shows clinical benefits in reasonable threshold probability with the nomogram (Fig. 3C).

Figure 2 Nomogram for diagnosis of benign and malignant liver lesions with AFP negativity.

Including age, ALT, LMR, Sex, HBV/HCV infected and single lesion. ALT, alanine aminotransferase; LMR, lymphocyte-to-monocyte ratio; HBV, hepatitis B virus; HCV, hepatitis C virus.

Figure 3 ROC curves, calibration curves and decision curves for the model.

(A) ROC curve of the nomogram in the training cohort. (B) Calibration curve of the nomogram in the training cohort. (C) Decision curve analysis for diagnosis in the training cohort. (D) ROC curve of the nomogram in the validation cohort. (E) Calibration curve of the nomogram in the validation cohort. (F) Decision curve analysis for diagnosis in validation cohort. (G) ROC curve of the nomogram in the external validation cohort. (H) Calibration curve of the nomogram in the external validation cohort. (I) Decision curve analysis for diagnosis in the external validation cohort.

We varified the nomogram to comfirm the reliability. In the validation cohort, AUC was about 0.882 (95% CI [0.815–0.949]) (Fig. 3D). The calibration curves also matched (Fig. 3E), and the DCA curves had good clinical practicability (Fig. 3F). Additionally, in the external validation cohort, AUC was about 0.811 (95% CI [0.687–0.935]) (Fig. 3G). The calibration curves and the DCA curves were as follows (Figs. 3H–3I). The model established in this study is confined to discriminating the benign or malignant nature of AFP-negative focal hepatic lesions and does not encompass specific pathological classifications.

Discussion

The distinction between the benign and malignant nature of AFP-negative focal hepatic lesions remains a substantial challenge in clinical research. This study endeavors to establish a reliable predictive tool for AFP-negative focal hepatic lesions, with the goal of enhancing in clinical triage and decision-making. Employing a systematic screening process, we identified clinically relevant and easily obtainable clinical characteristics and laboratory parameters related to AFP-negative focal hepatic lesions in primary healthcare settings. Ultimately, six risk factors were determined, including viral infection status, age, gender, ALT, solitary lesion status, and lymphocyte-to-monocyte ratio (LMR). A novel predictive model based on these six factors was successfully developed to forecast the benign or malignant nature of AFP-negative focal lesions, thereby enhancing diagnostic efficiency.

The predictive tool established in this study, incorporating the identified six factors, exhibits robust discriminatory capability in predicting the malignancy or benignity of AFP-negative focal hepatic lesions. It demonstrates high clinical utility and effectiveness. Compared to traditional imaging methods, this model serves as a low-cost and efficient non-invasive predictive tool, facilitating the prediction of malignancy or benignity in AFP-negative focal hepatic lesions. All incorporated indicators are relatively easily accessible across various healthcare facilities. In the development and validation cohorts of the model, satisfactory results were obtained, showcasing outstanding predictive accuracy and superior net benefit compared to traditional assessment models. The application of this strategy provides clinicians with additional information, aiding them in making more personalized clinical triage and treatment decisions. The prevalence of AFP-negative lesions makes accurate diagnosis critical. Our model addresses this need by providing a low-cost, effective tool for triaging AFP-negative HCC.

It is widely recognized that males over the age of 40 are at a high risk of developing liver cancer (Fan et al., 2020). The male gender is considered a high-risk factor for liver cancer, possibly due to the activation of androgen receptors, which may promote the growth and infiltration of liver cancer cells (Wu et al., 2023). Additionally, the inhibitory effects of endogenous estrogen metabolites on proliferation, promotion of apoptosis, and suppression of angiogenesis position estrogen as a positive factor in inhibiting tumor growth (Camporez et al., 2013). Gender is mentioned as a key factor in guides, and even though it does not perform well in the statistics, it was included in the model.

Epidemiological data indicate that as many as 77% of liver cancer patients in China have a history of hepatitis B virus (HBV) infection (Zhang et al., 2022). HBV carriers have a liver cancer incidence rate more than 100 times higher than non-carriers (Jia et al., 2015), highlighting the close association between liver cancer and HBV. HBV infection induces chronic inflammation in liver cells, leading to repeated proliferation and death of liver cells, thereby increasing the risk of developing liver cancer (Chen et al., 2006; Guo et al., 2000; Piciocchi et al., 2016). Furthermore, research suggests that 10–20% of chronic HCV-infected individuals will experience complications, with HCV increasing the risk of early onset cirrhosis and liver cancer in multiple-infected individuals (Liu, Liu & Liu, 2022).

Serum ALT primarily exists in the cytoplasm of liver cells and serves as a crucial indicator for assessing the degree of liver damage. The occurrence of liver cancer results in severe liver cell damage, leading to varying degrees of elevation in serum ALT levels in liver cancer patients (Zhang et al., 2019). Simultaneously, lymphocytes and monocytes play critical roles in the process of tumor occurrence and development. Although LMR is considered an important prognostic marker for solid tumor patients (Minici et al., 2022), its mechanism in predicting the prognosis of tumor patients is not entirely elucidated. Multiple studies (Mandaliya et al., 2019; Miyahara et al., 2020; Rajwa et al., 2018; Trinh et al., 2020) suggest that a decreased LMR before treatment is associated with unfavorable overall survival (OS) in various tumor subgroups, including HCC. Some related studies (Wu et al., 2016; Yang et al., 2018) indicate that LMR is an important predictive indicator for liver cancer.

This study identified predictive factors for the malignancy of AFP-negative focal hepatic lesions, with imaging factors including the presence of a single lesion. Previous studies have confirmed that AFP-negative HCC typically presents as a single lesion (Eisenbrey et al., 2021; Maruyama et al., 2021). Although advanced imaging techniques such as CT and MRI are currently applied in clinical practice for discriminating focal hepatic lesions, these methods are not ideal due to factors such as high cost, radiation exposure, and the influence of imaging physician experience on judgment results. In actual clinical practice, the assessment of imaging is often subject to the subjective factors of imaging physicians.

The clinical diagnosis of AFP-negative focal hepatic lesions has consistently presented challenges. Previous predictive models have primarily focused on comparative studies involving AFP-negative hepatocellular carcinoma patients and healthy individuals (Liu et al., 2023; Luo et al., 2019). However, these models have overlooked the potential confusion with other focal hepatic lesions that share similarities with AFP-negative hepatocellular carcinoma. In contrast, our study broadens its scope by analyzing various common focal hepatic lesions, rendering the established model more suitable for distinguishing malignant AFP-negative focal hepatic lesions in practical clinical applications. Moreover, the data utilized in this study are sourced from pre-surgery or pre-biopsy cross-sections, lending a clearer clinical significance to the predictive model in guiding surgical treatment strategies. The predictive model does not distinguish between types of pathology, but relies on non-specific indicators as triage tools for clinical diagnosis and treatment. Implementation of this simplified predictive model can significantly reduce diagnostic waiting times, address clinical diagnostic and treatment dilemmas, and also reduce patient anxiety and psychological burden. In addition, the variables used in the model are readily available, cost-effective, and easily accessible, making it a low-cost, highly efficient predictive tool that is applicable to a variety of settings, including primary healthcare providers and community health services, and adapted to China’s national conditions, with significant economic and social benefits.

However, the current study faces several limitations. Firstly, the retrospective design poses challenges in eliminating inherent biases. Secondly, the model lacks the inclusion of additional tumor biomarkers such as PIVKA-II (Mohamedein et al., 1995; Wang et al., 2022) and AFP-L3 (Tayob et al., 2023). This exclusion is primarily based on our model construction’s emphasis on economic feasibility, convenience, and applicability in primary healthcare institutions.While a previous study (Fotouhi et al., 2023) has highlighted the central role of imaging in liver lesion diagnosis, our model focused on easily accessible clinical parameters for cost-effectiveness. Future iterations could integrate imaging data to improve accuracy, particularly in distinguishing benign from malignant lesions, where clinical parameters alone may be insufficient. Additionally, our external validation cohort, derived from a single primary care institution (JingNing People’s Hospital), showed slight reductions in the ROC curve, calibration, and DCA compared to the training set due to differences in the source and composition of benign and malignant cases. Future research will involve larger, more diverse cohorts from multiple centers to enhance the model’s generalizability and robustness. Despite these limitations, the model demonstrated good external generalizability, suggesting its potential applicability. Further research with a larger sample size and independent prospective external validation is needed to confirm its accuracy.

In summary, we have amalgamated six variables, including medical history, imaging diagnosis, and laboratory indicators, to construct a predictive tool aimed at predicting the benign or malignant risk of AFP-negative hepatic focal lesions.

Supplemental Information

Supplemental Information 1 Raw data.

Supplemental Information 2 Patients with liver-occupying lesions who were AFP-negative were divided into training, validation, and external validation sets.

Independent risk factors, including age, gender, ALT levels, HBV/HCV infection status, LMR, and single lesion presence, were identified to construct a predictive model. This model aims to reduce the 3-6 month waiting time for follow-up and facilitate rapid clinical decision-making.

We sincerely appreciate all the patients and healthy individuals who participated in this study.

Additional Information and Declarations

Competing Interests

The authors declare that they have no competing interests.

Author Contributions

Yu Lin conceived and designed the experiments, performed the experiments, authored or reviewed drafts of the article, and approved the final draft.

Qianyi Wang performed the experiments, analyzed the data, prepared figures and/or tables, and approved the final draft.

Minxuan Feng analyzed the data, prepared figures and/or tables, and approved the final draft.

Jize Lao analyzed the data, prepared figures and/or tables, and approved the final draft.

Changmeng Wu analyzed the data, prepared figures and/or tables, and approved the final draft.

Houlong Luo analyzed the data, prepared figures and/or tables, and approved the final draft.

Ling Ji conceived and designed the experiments, authored or reviewed drafts of the article, and approved the final draft.

Yong Xia conceived and designed the experiments, authored or reviewed drafts of the article, and approved the final draft.

Human Ethics

The following information was supplied relating to ethical approvals (i.e., approving body and any reference numbers):

The study was conducted in accordance with the revised 2013 Declaration of Helsinki and received approval from the Ethics Committees of Peking University Shenzhen Hospital (Approval [Research] [2023] No. 075) and JingNing People’s Hospital, Gansu Province (Approval No. 2023101501).

Data Availability

The following information was supplied regarding data availability:

The raw data is available in the Supplemental File.

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
