# Peer review of "A cost-effective predictive tool for AFP-negative focal hepatic lesions of retrospective study: enhancing clinical triage and decision-making"

_PeerJ, doi:10.7717/peerj.19150_

## Round 0.1 · original submission · Major Revisions

Dear authors,

Thank you for your submission addressing the critical diagnostic gap in AFP-negative hepatocellular carcinoma (HCC). While the study presents robust data and a well-constructed diagnostic model, several critical issues need to be addressed. Some of the highlighted revisions encompass: the limitation of using a single hospital cohort (JingNing People's Hospital) which should be emphasized or addressed more explicitly, as noted by Reviewer 1. Broader validation with diverse populations would strengthen the study's applicability. Also, the need to detail the clinical features of AFP-negative HCC in the introduction is crucial for contextualizing the study's significance and relevance. Clarify whether imaging data were used, justify its exclusion, or discuss its potential integration in future models, and provide details on the proportion of AFP-negative HCC cases linked to viral hepatitis, given its high prevalence in China. This will further assess the model's applicability. A minor note: consistently expand abbreviations upon first use, including in the abstract, to ensure clarity for a broader readership.

Refer to the reviewers' comments for further details.

I look forward to receiving your revised manuscript.

Reviewer 1 ·

Basic reporting

no comment

Experimental design

The external cohort consists of 78 patients from a single hospital (JingNing People's Hospital in Gansu Province), which might not be representative of the broader population. A larger, more diverse external cohort would help to validate so this can be emphasized more in limitations.

Validity of the findings

no comment

Additional comments

Good study in general looking up at a valid score development

Reviewer 2 ·

Basic reporting

The paper effectively communicates its entirety in English, the references offer ample domain-specific context, and the charts and graphs accurately depict the research findings.

Experimental design

1.In the Materials and Methods section, the authors should explicitly define the specific assay employed for AFP detection and delineate the threshold level below which a diagnosis of AFP-negative HCC can be established.
2.I am interested in determining the proportion of AFP-negative HCC cases in the paper that are attributable to viral hepatitis, especially considering that China has the world's largest viral hepatitis patient population. This information is crucial for assessing the applicability of the model presented in the paper.

Validity of the findings

The paper provides a wealth of detailed data, employs sound statistical methods, and presents the findings appropriately, thereby holding significant value in aiding the clinical diagnosis of AFP-negative HCC.

Additional comments

Would it be possible for the authors to elucidate the clinical features of AFP-negative HCC within the introduction, thereby enabling readers to attain a fuller grasp of the paper’s context?

Reviewer 3 ·

Basic reporting

This study addresses the significant challenge of diagnosing alpha-fetoprotein (AFP)-negative hepatic lesions, focusing on developing a cost-effective predictive tool for clinical triage and decision-making. Using retrospective data from three cohorts, the authors constructed and validated a diagnostic model based on multivariate logistic regression. The tool demonstrated robust diagnostic performance, with high area-under-the-curve (AUC) values, making it a potentially valuable addition to clinical practice.
The authors should expand abbreviations the first time they appear in the manuscript, including the abstract. For example, terms like "AFP," "ALT," and "LMR" are used without prior explanation, which could hinder understanding for readers unfamiliar with these terms.

Experimental design

The study lacks clarity on whether imaging data were included in the predictive model. Given that imaging plays a central role in diagnosing liver lesions, excluding it from the model would represent a significant limitation. If imaging data were used, this should be explicitly stated and discussed in the methods. If not, the authors should justify its exclusion and consider incorporating it in future iterations of the model: https://pubmed.ncbi.nlm.nih.gov/37964787/ .

Validity of the findings

The manuscript should better emphasize that AFP-negative lesions are very common in cirrhotic patients and often constitute the majority of hepatocellular carcinomas (HCCs). This point would add context to the significance of the tool and its relevance to routine clinical practice.

The authors could discuss the potential for integrating this tool with imaging-based diagnostic methods to further enhance its predictive power. This integration might address limitations in distinguishing benign and malignant lesions, especially in cases where clinical parameters alone are insufficient.

Additional comments

The manuscript is a commendable effort to address a critical diagnostic gap. With improvements in abbreviation usage, a clearer discussion of imaging data, and a more detailed consideration of the prevalence of AFP-negative HCCs, the paper could achieve greater clarity and impact.

---

## Round 0.2 · accepted · Accept

Dear authors,

Thank you for your re-submission and diligence with your manuscript. My sincere condolences over the passing of Qianyi Wang.

The reviewers approved of your revisions. Your manuscript is now ready for publication. Many thanks and congratulations.

Reviewer 2 ·

Basic reporting

no comment.

Experimental design

no comment.

Validity of the findings

no comment.

Additional comments

no comment.

Reviewer 3 ·

Basic reporting

-

Experimental design

-

Validity of the findings

-

Additional comments

-